# The Energy Characteristics of the Surface of Statistical Copolymers

**DOI:** 10.3390/polym15081939

**Published:** 2023-04-19

**Authors:** Anatoly E. Chalykh, Valentina Y. Stepanenko, Tatiana F. Petrova, Anna A. Shcherbina

**Affiliations:** 1Frumkin Institute of Physical Chemistry and Electrochemistry, Russian Academy of Sciences (IPCE RAS), bld.4 Leninsky Prospect, 119071 Moscow, Russia; 2Advanced Engineering School of Chemical Engineering and Machinery, Mendeleev University of Chemical Technology, 9 Miusskaya Square, 125047 Moscow, Russia

**Keywords:** statistical copolymers, elastomeric materials, surface energy, adhesives, boundary wetting angles, conformational rearrangements of macromolecules, activation energy of α- and β-transitions

## Abstract

The results of systematic studies on the surface energy *γ* and its polar *γ*^P^ and dispersion *γ*^D^ components of statistical copolymers of styrene and butadiene, acrylonitrile and butadiene, and butyl acrylate and vinyl acetate, with regard to their thermal prehistory, are generalized. Along with copolymers, the surfaces of their composing homopolymers were examined. We obtained the energy characteristics of the adhesive surfaces of copolymers that contacted with air, high-energy aluminium *Al* (*γ* = 160 mJ/m^2^), and the low-energy substrate surface of polytetrafluoroethylene F4 (PTFE) (*γ* = 18 mJ/m^2^). The surfaces of copolymers in contact with air, aluminium, and PTFE were investigated for the first time. It was found that the surface energy of these copolymers tended to occupy an intermediate value between the surface energy of the homopolymers. The additive nature of the change in the surface energy of the copolymers with their composition, as previously established in the works of Wu, extends to the dispersive component of the free surface energy *γ^D^* and the critical surface energy *γ_cr_*, according to Zisman. It was shown that a significant influence on the adhesive activity of copolymers was exerted by the substrate surface upon which the adhesive was formed. Thus, for the butadiene–nitrile copolymer (BNC) samples formed in contact with a high-energy substrate, their surface energy growth was associated with a significant increase in the polar component of the surface energy *γ^P^* from 2 mJ/m^2^ for the samples formed in contact with air, to an increase from 10 to 11 mJ/m^2^ for the samples formed in contact with *Al*. The reason why the interface influenced the change in the energy characteristics of the adhesives was the selective interaction of each macromolecule fragment with the active centres of the substrate surface. As a result, the composition of the boundary layer changed and it became enriched with one of the components. The structure of such layers is nonequilibrium. The thermal annealing of copolymers in the mode of a stepwise temperature increase led to a convergence in the values of *γ*, asymptotically tending to the value characteristic of the surface of the copolymers formed in air. The activation energies for the processes of the conformational rearrangements of the macromolecules in the surface layers of the copolymers were calculated. It was found that the conformational rearrangements of the macromolecules in the surface layers occurred as a result of the internal rotation of the functional groups that determined the polar component of the surface energy.

## 1. Introduction

Statistical copolymers are known to occupy a special place in the material science of adhesive compounds [1,2,3,4]. In recent years, the synthesis of pressure-sensitive adhesives has been developed most intensively [5]. Traditionally, this class of adhesive is developed based on cis-polyisoprene copolymers, butadiene–styrene copolymer (BSC) and butadiene–nitrile copolymer (BNC), polyisobutylene (PIB) and its copolymers, butadiene and isoprene block copolymers, and acrylic and polyurethane polymers and copolymers. The latter constitute a large group of structural adhesives, in particular, impact-active acrylates and cyanoacrylates, as compositions with numerous additives, including pharmaceuticals in transdermal systems [6,7]. As a reminder, this class of adhesive requires a sufficiently high fluidity and tackiness so that the required bond strength is realised during the initial contacting phase of the adhesive system elements when they are pressed together. This characteristic of pressure-sensitive adhesives determines their wide use in the production and application of adhesive tapes, transdermal materials, self-adhesive labels, decorative finishes in vehicles, and aerospace vehicle assembly technology [8]. Up until now, investigations into the adhesive properties of nitrile, butadiene styrene, and acrylic copolymers have mainly concerned the determination of the strength of their adhesive bonds with rubbers, elastomer vulcanizates, polyolefins, and metals [9]. The energy characteristics of copolymers are practically not investigated and the thermodynamic work of their adhesion is not estimated, which complicates the interpretation of the mechanism of the adhesion interaction in these systems.

Other adhesive compositions that are widely used in hot melt adhesives are based on the copolymers of ethylene and vinyl acetate, ethylene and propylene, and polyamide copolymers [10]. These adhesives are characterized by their excellent flexibility, high-impact toughness, low fatigue, and ability to retain their operational properties in the temperature range from 250 to 350 K. Their chemical, radiation-chemical, and composite modifications can increase the strength properties of adhesive bonding [11].

It has been shown in [12,13] that the surface activity of statistical copolymers is traditionally characterised by their surface energy *γ* and its polar *γ^P^* and dispersion *γ^D^* components. These characteristics change with the nature of the contact surface: the copolymer composition and molecular weight, the block length, the phase structure, and the microphase separation of the monomers. We note that information about the energy characteristics of substrate and adhesive surfaces underlies modern ideas about the state of interfacial boundary layers and the thermodynamics of polymer mixing; they are important for predicting and interpreting the strength of adhesive compounds, choosing surface modification methods, and interpreting the mechanism of adhesive interactions [14,15]. Nevertheless, research within this field of polymer, composite, and material physics is sketchy, fragmentary, and random. Their comparative analysis is difficult because studies use different substrates and methods for synthesising the monolayers and assessing the strength of the adhesive bonds.

The aim of this work is to deepen our understanding of the state of interfacial boundary layers, the thermodynamics of polymer mixing, the improvement of adhesive bonding strength, the choice of surface modification methods, and the study of the mechanism of adhesive interactions, based on a systematic study of the surface energies of statistical copolymers of different compositions and thermal prehistories. The influences of the nature of the contacting surface and the functional groups of different polarities were evaluated.

## 2. Objects and Methods of Investigation

Traditional elastomeric materials based on amorphous carbonaceous statistical copolymers of various natures and compositions, such as the butadiene styrene rubbers BSC-10, BSC-18, BSC-28, and BSC-40 and copolymers of styrene and butadiene (prototypes) with a styrene content from 10 to 92 wt%—BSC (VNIISK, Saint-Petersburg, Russia), were used as the objects of investigation. The butadiene–nitrile rubbers BNC-18, BNC-26, BNC-40, and BNC-50 and copolymers of acrylonitrile and butadiene with an acrylonitrile content from 18 to 65 wt%, as well as copolymers of butyl acrylate and vinyl acetate (BAC) and homopolymer polybutyl acrylate (PBA) (JSC «NII Polimery», Joint Stock Company “Academician V.A. Kargin Research Institute of Chemistry and Technology of Polymers with Pilot Plant ”Dzerzhinsk, Russia) were also investigated. Additionally, the energetic characteristics of homopolymers—polystyrene (PS), polyacrylonitrile (PAN), polybutadiene (PBD), and polyvinyl acetate (PVA) (VNIISK, Saint-Petersburg, Russia) were studied. The relative permittivities of the monomer links were 2.0, 2.0 for butadiene, 2.5 for polystyrene, 3.5 for polyacrylonitrile, 3.22 for polyvinyl acetate, and 2.1 for polybutyl acrylate. The characteristics of the homopolymers are given in Table 1. The compositions of copolymers were identified by the glass transition temperature (*T_g_*) of the homopolymers and copolymers, using the DSC method [16] on thermal analyzer “DSC 204 F1 Phoenix” (“Netzsch”, Selb, Germany) on the copolymer films formed from the solutions on the PTFE substrate.

The surface energy characteristics of the statistical copolymers and homopolymers were determined for the two groups of the film adhesives as a part of the model adhesion samples. The first group consisted of initial polymer films that were from 100 to 160 µm thick and 30 × 20 mm in area, prepared from polymer melts and solutions in different solvents (toluene, tetrahydrofuran). The second group consisted of the same polymers, but heat treated.

When the model polymer adhesion samples were obtained from the melts, a suspension was placed on aluminium foil and/or the PTFE film attached to a clamping table. After melting at 403 K for 30 min, the polymer was duplicated with another foil, squeezing the clamp until films of the given thickness were obtained. The model adhesion compounds that were thus obtained were then annealed at temperatures between 353 and 403 K for 30 min, followed by being air cooled and quenched in liquid nitrogen.

Special cuvettes were used to obtain the model adhesion samples of the films viairrigation on the surface *Al* and PTFE substrates. Solutions with a concentration of 5 wt% of toluene or tetrahydrofuran were prepared. The drying process consisted of two stages. Initially, the cuvette mass was brought to a constant value at 298 K for 5 days, with a residual solvent content from 10 to 12 wt%. The films were then vacuum dried at 353 K for 5 h. According to a mass thermal analysis [17], the samples obtained this way contained 1 wt% of residual solvent. The further heat treatment of the model adhesion samples did not differ from that described above.

The surfaces of the polymer films after contact with air, aluminium, and PTFE were studied. Air is an energetically inert medium, whereas for aluminium, the surface energy *γ* = 160 mJ/m^2^ (high-energy substrate surface), and for PTFE, *γ* = 18 mJ/m^2^ (low-energy surface). Special attention was directed to the destruction of the adhesion bonds formed in contact with *Al* and PTFE. For this purpose, drops of ethyl alcohol solution with a concentration of 40 wt% were introduced into the splitting zone, which ensured the adhesive character of the fracture, with minimal plastic deformation of the adhesives. The morphological structure of the degradation zone was identified by scanning electron microscopy and an X-ray microanalysis on a JSM 6060A firm “JEOL” (Japan) microscope, with 15 keV accelerating voltage and a beam current of 10^−9^ amp.

The measurements of the boundary wetting angles (*θ*) of the copolymer and homopolymer surfaces were performed on Easy Drop (Germany) at a temperature of 293 ± 1 K. The characteristics of the test liquids are given in Table 2.

The methodology of measurements of *θ* and cos *θ* and the calculations of *γ*, *γ^D^*, and *γ^P^* did not differ from the methods described in [18]. For all the investigated samples in the coordinates of the Fawkes equation (1 + *cos θ*) − (*γ*^1/2^/*γ*), there was a linear dependence with a correlation coefficient that was not lower than 0.97.

## 3. Results and Discussion

The copolymers studied are classified as flexible-chain polymers (the Cohn segment length is from 2.0 to 2.2 nm). Their glass transition temperatures vary in the range from 203 K for polybutadiene to 357 K for PS and 373 K for PAN (Table 1). According to the DSC data, the transition of the copolymers into the glassy state at room temperature, was observed with the content of the styrene and acrylonitrile links being about 65 wt%. Vinyl acetate and butyl acrylate copolymers are in a highly elastic state under these conditions. It should be noted that, along with monomer links, the structural element that affects the surface energy of the polymers is the backbone of the macromolecular chains, which is similar to polyolefins in its contribution to the energy characteristics [19,20].

Figure 1a, Figure 2a and Figure 3a show the typical surface energy dependencies of the homopolymers (PB, PS, PAN, PVA, and PBA) and the copolymers of BSC, BNC, and BAC, which were formed in contact with air. 

Firstly, it is notable that the surface energy of copolymers tends to occupy an intermediate value between the surface energy of homopolymers. This result applies to the polar BNC, BAC, and non-polar BSC adhesives and corresponds to the data previously obtained by Wu [13].

On the secondary side, the surface energy dependence of the studied class of statistical copolymers from their composition obeys (Figure 1) to the additive equation:(1)$$\gamma =x_{1}\times \gamma_{1}+x_{2}\times \gamma_{2},$$
where *x*_1_ and *x*_2_ are the mass fractions of the surface occupied by the monomer links of the homopolymers, with surface energies *γ*_1_ for PB and *γ*_2_ for PS or PAN. 

It is interesting to note that the additive nature of the change in the surface energy of copolymers with their composition also applies to the values of the critical energy *γ_cr_* obtained by Zisman [5] (see Table 3):(2)$$\gamma^{D}=\gamma_{1}^{D}\times x_{1}+\gamma_{2}^{D}$$

This dependence is less clear for the polar component of the surface energy, although, even in this case, we can talk about its tendency to increase with the growth of the styrene, acrylonitrile, and vinyl acetate segments in the macromolecules of copolymers.

Thirdly, following the ideology of Cassie and Baxter [21], we estimated the surface fractions *x*_1_ occupied by the monomer links of PS, PAN, and PVA. For this purpose, we used the experimental data on the surface energies of the BSC and BNS copolymers in Equation (1).

Figure 4 shows the correlation dependence of the mass fraction of the surface occupied by butadiene links on the composition of the BSC and BNS copolymers in De Boer coordinates [22].

A well-defined linear dependence between the values of *x*_2_ estimated and *x*_2_ obtained experimentally, with correlation coefficients of 0.97 allows us to suggest, that the surface composition of the adhesives formed in contact with air, is identical to that of the bulk phase of the copolymers. Thus, we can assume that the conformations of the macromolecules of the copolymers in the volume and surface layers are identical, and the main contribution to the adhesive activity of these copolymers is made by a dense macromolecular monolayer.

We estimated additional information about the contributions of the polar and dispersion components of the free surface energy using the Schonhorn method [23], i.e., using the ratios *γ^D^*/*γ* and *γ^P^*/*γ*. For BSC, this ratio had the numerical values *γ^D^*/*γ* = 0.92 and for BNC, *γ^D^*/*γ* = 0.83. It is interesting to note that the proportion of the copolymer surface that was occupied by the dispersion component elements *γ^D^* was closely similar for both the copolymers, whereas the proportion of the surface occupied by the polar component *γ^P^* was significantly different: for BSC *γ^P^*/*γ* = 0.06 and for BNC *γ^P^*/*γ* = 0.19, which ultimately determined their surface and adhesive activity. We further assumed that the proportion of the polar groups in the macromolecules of the copolymers was proportional to the ratio *γ^P^*/*γ* [23].

A specific feature of these BSC and BNC copolymers is related to the temperature conditions of the adhesive surface formation in the glass transition region of the copolymers. We can see (Figure 1) that, in this case, in the region of the compositions close to the homopolymers PS and PAN, there is a scatter in the values of the surface energy and its components that are associated with the conditions of obtaining copolymer samples, but this did not exceed the values of the surface energy characteristics of the homopolymers: for PS *γ* = 40 ± 2 mJ/m^2^ and PAN *γ* = 50 ± 2 mJ/m^2^. It is interesting to note that, for butyl acetate copolymers, this effect was not observed.

It was found that the adhesive activity of the copolymers was significantly affected by the substrate surface upon which the adhesive layer was formed. As an example, Figure 1b, Figure 2b and Figure 3b show the concentration dependences *γ*, *γ^P^*, and *γ^D^* for the BSC, BNC, and BAC films after their separation from the high-energy (*Al*) and low-energy (PTFE) surfaces. It is not able that the concentration dependences of the copolymer surface energies for the different substrates were some what different, due to the different compositions and structural organizations of the surface layers. Thus, for the samples formed in contact with a high-energy substrate, an increase in the surface energy of the copolymers and homopolymers was observed. For BNC, this effect was associated with a significant increase in the polar component of the surface energy, from *γ^P^* = 2 mJ/m^2^ for the samples formed in contact with air, to *γ^P^* from 10 to 11 mJ/m^2^ for the samples formed in contact with *Al*. For the samples formed in contact with PTFE, *γ^P^* was of the order of 1.8 mJ/m^2^. The other values were situated between the two extremes of *γ^P^*, which is typical for PVA (*γ^P^* = 12.0 mJ/m^2^) and PE (*γ^P^* = 1.1 mJ/m^2^). It is interesting to note that the dispersion component of the surface energy, which describes the packing density of the monomer units in the surface layers of copolymers, varied slightly for BNC.

A different pattern was observed in the case of the non-polar BSC copolymers. For these copolymers, the change in the total surface energy was related to the change of its dispersion component *γ^D^*/*γ*, from 0.91 for the samples formed in contact with air, to 0.93 for those formed in contact with *Al*. At the same time, the polar component of BSC has hardly changed. Thus, one can suppose that the data obtained testify to the enrichment of the surface layer with the polar functional groups of acrylonitrile in the case of BNC, and to the formation of a dense layer of styrene units due to the orienting effect of the high-energy substrate in the case of BSC. It can be assumed that the acrylonitrile and styrene links at the contact of the BNC and BSC with the high-energy surface move from the volume of the macromolecule tangle to its surface layer, the "pubescence". Ultimately this has an influence on the macroscopic characteristics of the adhesives (Figure 1b, Figure 2b and Figure 3b).

For all the copolymers, as the number of active functional groups grew, the energy parameters of the adhesives increased, asymptotically approaching some limiting value close to the surface energy of the corresponding PAN or PVA homopolymers. To describe these isotherms, we used the traditional Langmuir equation [24]:(3)$$\frac{\gamma^{P}}{\gamma }=K\times x_{2}/(1+K\times x_{2})$$
where *x*_2_ is the copolymer composition, *K* is the group adsorption equilibrium constant, and *γ^P^*/*γ* is the polar component of the surface energy, proportional to the degree of the filling of the polar groups in the substrate. It can be seen (Figure 5) that the dependence plots *γ*/*γ^p^* − *1*/*x*_2_ area linear relationship, cutting off a segment that is proportional to the surface concentration of the active centres on the substrate surface of the ordinate axis, and the slope of these straight lines allows for the determination of Langmuir equation constant.

In the future, we intend to use these values for determining the thermodynamic parameters of the process of forming the adhesion bond.

A different pattern was observed in the case of the adhesion formation in contact with the low-energy surface of the PTFE substrate. For these samples (Figure 1b, Figure 2b and Figure 3b), the surface energy of the homopolymers and copolymers was 10% lower than that of the equilibrium surface energy formed in air. This also applied to the surface energy components *γ^D^* and *γ^P^*. Based on the ratio *γ^D^*/*γ*, we can assume that, in this case, a “loose” surface layer was formed, which was practically devoid of polar functional groups. For both types of the BSC and BNC copolymers, the surface energy after the formation on the PTFE substrate had extremely low values, which were lower when formed in air. This pattern was observed for both the polar and dispersive components. It can be assumed that, in this case, the acrylonitrile and styrene links, under the influence of repulsive forces, occupied the position inside the tangle of the adhesive macromolecule, exposing the butadiene links and the copolymer chain backbone. This effect has been previously described as an example for epoxy oligomers with different molecular weights [23]. The selectivity of the interaction of each macromolecule fragment with the surface active centres is a result of the influence of the energy characteristics. The consequence of this is a change in the composition of the boundary layer and its enrichment with one of the components. Previously, such effects were observed for block copolymers and polymer mixtures [21], and were associated with the microphase separation of the adhesive. In our case, the selective enrichment of the surface layer was not accompanied by a phase separation of the copolymers, but led to the formation of structurally gradient surface layers. Previously, it was assumed that balls of macromolecules would unfold on the contact surface, forming folded structures. It was found that, in the surface layer of a vinyl chloride and vinyl acetate copolymer formed on a high-energy substrate, a decrease in the concentration of the acetate polar groups was observed with time exposed to air. This process is more active with an increasing temperature. The dependence of the boundary layer composition on the conditions of the adhesion bond formation has been shown in the examples of the following systems: a statistical copolymer of ethylene and methacrylic acid–polyethylene; a graft copolymer of maleic anhydride (2.3%) to polyethylene–polyethylene; and a graft copolymer of vinyltrimethoxysilane (1.2%) to a copolymer of ethylene and vinyl acetate (72:28)—polyethylene. It also seems natural to assume that, in the case of low-energy substrates in the surface layers of copolymers, the functional groups are reoriented so that they are directed inside the tangle volume. The authors refer the calculated surface energy to the chain backbone.

In conclusion, we note that the structure of such layers is nonequilibrium. The thermal annealing of copolymers in the mode of a stepwise temperature increase, as shown in Figure 6, leads to a gradual convergence of the values for *γ*, asymptotically tending to the value characteristic for the surface of copolymers formed in air (marked in the figure with a dotted line). The energy characteristics change more intensively within in the temperature ranges for the α- and β-transitions of copolymers.

Figure 6 shows typical results from measurements related to the relaxation in the surface characteristics of copolymers. It can be seen that, for practically all the copolymers, the same trend of changes in the surface energy and its components with annealing time is observed. It is interesting to note that, for the samples formed in contact with the high-energy surface, the *γ* and *γ^P^* decrease, whereas, for the samples obtained on the PTFE surface, growth is observed.

The kinetic relaxation curves were processed within a single-barrier transition with one relaxation time *τ* [25]:(4)$$\frac{\gamma -\gamma_{\infty }}{\gamma_{0}-\gamma_{\infty }}=exp\left(-\frac{t}{\tau }\right)$$where $\gamma_{0}$ and $\gamma_{\infty }$ are the surface energy at zero and an infinitely large time, τ is the relaxation time, and t is the time.

It has been shown for the first time that the relaxation kinetics of *γ* and *γ^p^* are satisfactorily described by this Equation (Figure 7), and that the relaxation times calculated for the polar component and the surface energy, in general, practically coincide with each other within an error. The activation energies of the processes of the conformational rearrangements of the macromolecules in the surface layers of the copolymers were calculated using these values. The numerical values of the activation energies of these structural rearrangements are given in Table 4.

The comparison of the presented values with the activation energies of the α- and β-transitions for the copolymers, which were obtained independently by methods of relaxation spectroscopy [25], suggests that the processes of the conformational rearrangements in the surface layers are responsible for the internal rotation of the functional groups that determine the polar component of the surface tension. If one accepts the view stated in [25], where the β-transitions are associated with the mobility of the side group of the copolymers, then one can assume that the conformational changes in the surface layers of the adhesives are associated with the rotation, that is, with their transition to volume. This means that the rotations of the side group, relative to the chain backbone, do not depend upon their position in space, but are determined by the internal rotation potentials, that is, they are the characteristic value for a given polymer.

## 4. Conclusions

It has been established that the adhesive activity of elastomers is considerably affected by the substrate surface upon which the adhesive is formed. The reason behind this interface influence on the change in the energy characteristics of the adhesives is the selectivity of the interaction of each macromolecule fragment with the active centres of the solid surface. As a result, the composition and density of the boundary layer are changed and it is enriched with one of the components. It was found that the structure of such layers is nonequilibrium. When the copolymers are thermally annealed in a stepwise temperature increase regime, the values of γ asymptotically approach the values that are characteristic of the surface of copolymers that are formed in air. The activation energies of the conformational rearrangements of the macromolecules in the surface layers of the copolymers have been calculated. It has been shown for the first time that the processes of the conformational rearrangements in these surface layers are responsible for the internal rotation of the functional groups that determine the polar component of the surface tension.

## Figures and Tables

**Figure 1 polymers-15-01939-f001:**
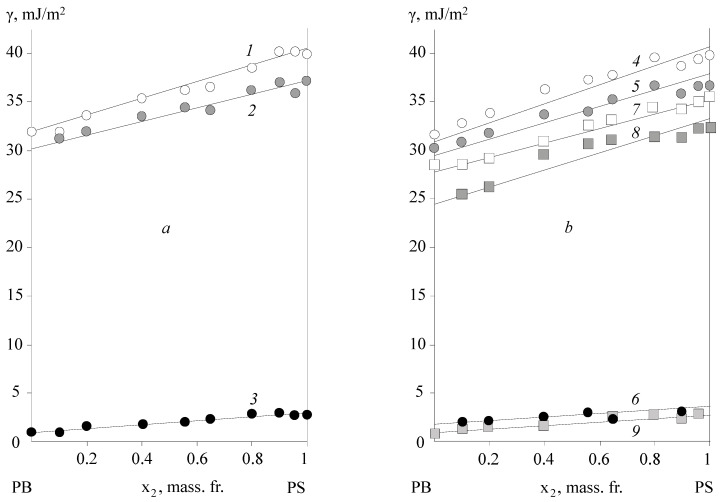
Dependence of the surface energy *γ* (1, 4, and 7) and its polar *γ^P^* (2, 5, and 8) and dispersion components *γ^D^* (3, 6, and 9) on the composition of BSC copolymers at 293 ± 1 K. BSC surface is formed on the: (**a**) air—(1, 2, and 3), and (**b**) aluminium—(4, 5, and 6) and PTFE—(7, 8, and 9).

**Figure 2 polymers-15-01939-f002:**
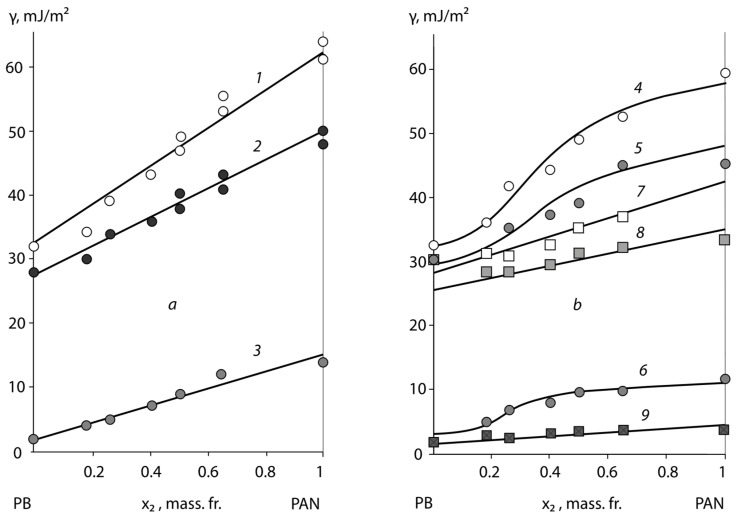
Dependence of the surface energy *γ* (1, 4, and 7) and its polar *γ^P^* (2, 5, and 8) and dispersion components *γ^D^* (3, 6, and 9) on the composition of BNC copolymers at 293 ± 1 K. BNC surface is formed on the: (**a**) air—(1, 2, and 3), and (**b**) aluminium—(4, 5,and 6) and PTFE—(7, 8, and 9).

**Figure 3 polymers-15-01939-f003:**
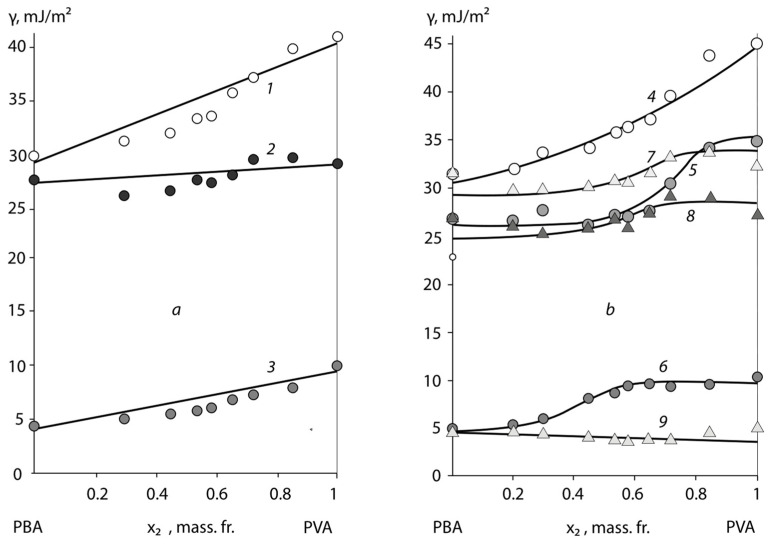
Dependence of the surface energy *γ* (1, 4, and 7) andits polar *γ^P^* (2, 5, and 8)and dispersion components *γ^D^* (3, 6, and 9)on the composition of BAC copolymers at 293 ± 1 K. BAC surface is formed on the: (**a**) air—(1, 2, and 3), and (**b**) aluminium—(4, 5, and 6) and PTFE—(7, 8, and 9).

**Figure 4 polymers-15-01939-f004:**
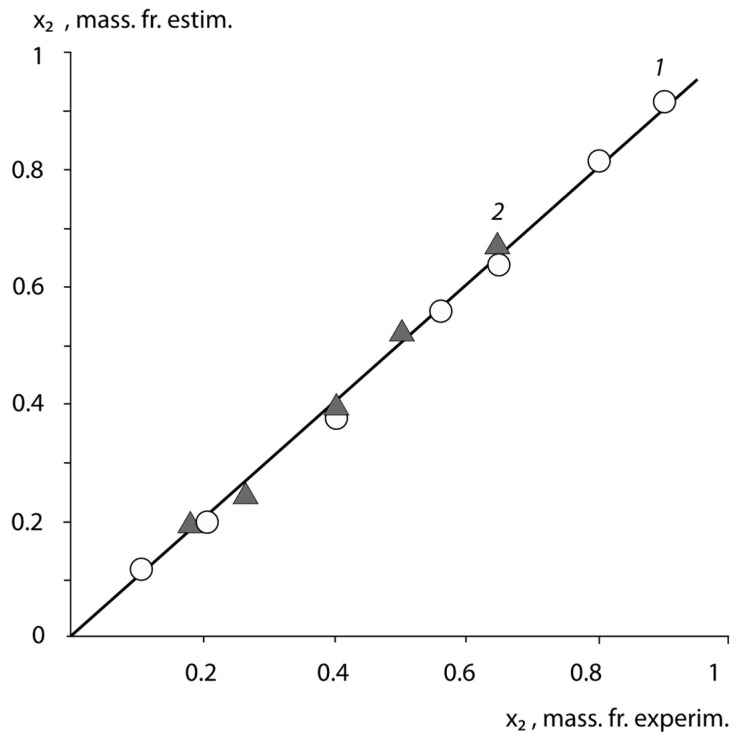
Correlation dependence of the composition of BSC (1), and BNC (2) copolymers (*x*_2_-content of styrene and acrylonitrile units) in De Boer coordinates.

**Figure 5 polymers-15-01939-f005:**
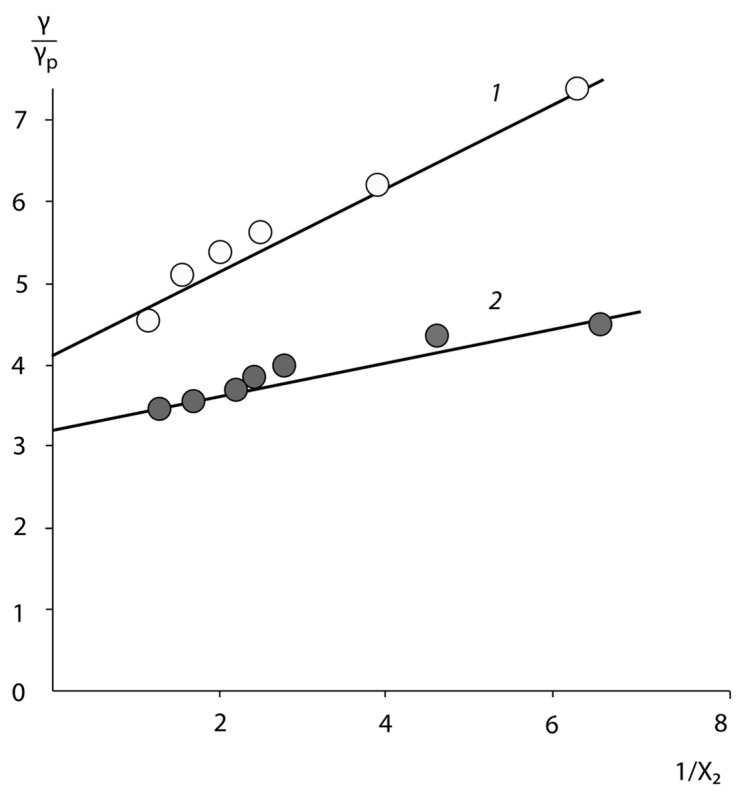
Isotherms of sorption by the polar groups of BNC (1), and vinyl acetate (2) on the *Al* substrate surface in coordinates with the linear version of the Langmuir equation.

**Figure 6 polymers-15-01939-f006:**
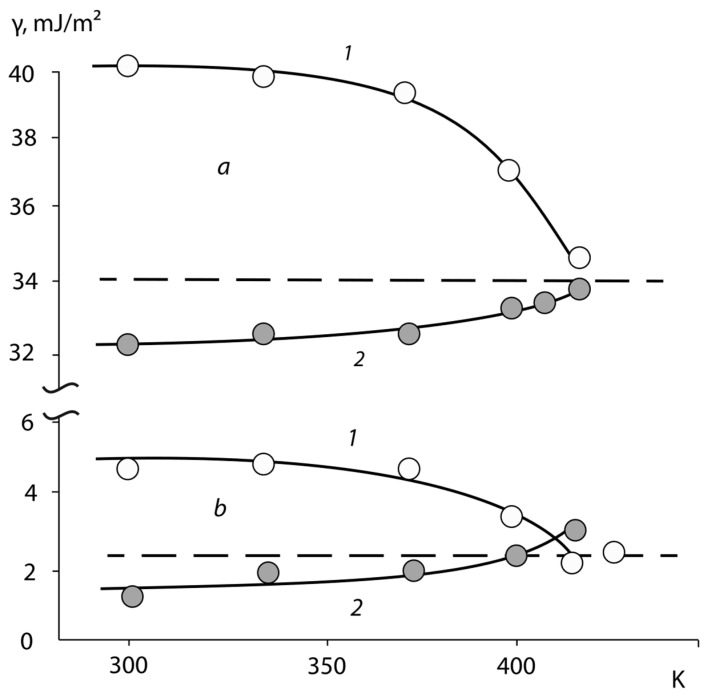
Dependencies *γ* (**a**), and *γ*^D^ (**b**) from annealing temperature of vinyl acetate and butyl acrylate copolymers (BAC-50). The copolymer surface is formed in contact with: *Al* (1), and PTFE (2).

**Figure 7 polymers-15-01939-f007:**
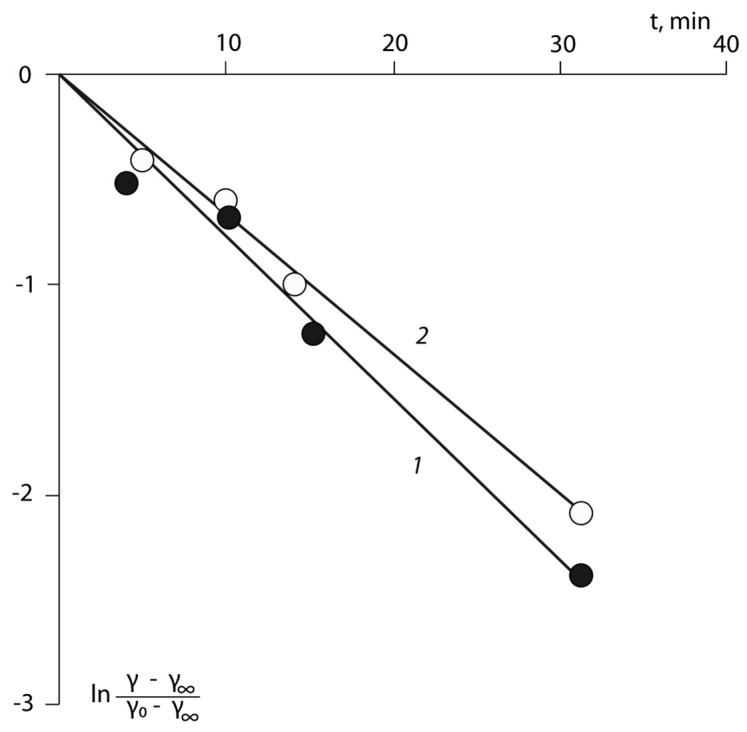
Relaxation dependencies γ for BNC-26 in coordinates of Equation (4). Surfaces are formed on the: 1—aluminium, and 2—PTFE.

**Table 1 polymers-15-01939-t001:** Characteristics of copolymers and homopolymers.

Subjects	M_ղ_ × 10^−5^_,_ g/mol *	Glass Transition Temperature *T_g_*, K	The Phase State at 293 K
BSC-(from 10 to 92 wt% styrene)	1.50	From 195 to 265	Amorphous
BNC-(from 18 to 65 wt% acrylic)	1.80	From 223 to 283	Amorphous
BAC	0.04	From 233 to 288	Amorphous, highly elastic
PS	1.05	378	Amorphous, glassy
PBA	0.56	228	Amorphous, highly elastic
PVA	0.05	302	Amorphous, glassy
PAN	1.80	380	Amorphous, glassy
PB	1.20	205	Amorphous, highly elastic

(*) according to capillary viscometry of copolymer and polymer solutions in toluene and tetrahydrofuran (THF).

**Table 2 polymers-15-01939-t002:** Characteristics of the test liquids.

№	Liquid	Density, g/cm^3^	Boiling Point, K	*γ^P^_lv_*, mJ/m^2^	*γ^D^_lv_*, mJ/m^2^	*γ_lv_*, mJ/m^2^
1	Water	1.00	373.0	50.2	22.0	72.2
2	Glycerin	1.260	563.0	30.0	34.0	64.0
3	Formamide	1.133	583.0	26.0	32.3	58.3
4	Methyleneiodide	3.325	454.0	2.3	48.5	50.8
5	Ethyleneglycol	1.109	570.2	19.0	29.3	48.3
6	1-Bromom-naphthalene	1.488	554.0	0.0	44.6	44.6
7	Dimethylsulfoxide	1.096	462.0	8.7	34.9	43.6
8	Aniline	0.022	457.4	2.0	41.2	43.2
9	Tricresyl phosphate	1.165	536.0	4.5	36.2	40.7

**Table 3 polymers-15-01939-t003:** Critical surface energies of copolymers according to Zisman.

Composition	*γ_cr_*_,_ mJ/m^2^	*γ_cr_* Calculated, mJ/m^2^
PB	32	33.5
BSC-10	33	33.8
BSC-20	35	34.6
BSC-40	36	36.0
BSC-55	37	37.0
PS	38	39.0

**Table 4 polymers-15-01939-t004:** Relaxation activation energies *γ* and *γ^p^* and activation energies, α- and β-transitions of copolymers [25].

Copolymer	E *, kJ/mol	E^P^ *, kJ/mol	E^β^, kJ/mol	E^α^, kJ/mol
BNC-18	42	47	37	61
BNC-26	45	46	43	63
BNC-40	49	48	46	66

(*) Data presented by the authors Bartenev G.M., Barteneva A.G. [25,26].

## Data Availability

The study did not report any data.

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
