# Peer review of "The Energy Characteristics of the Surface of Statistical Copolymers"

_polymers, 2023, doi:10.3390/polym15081939_

Round 1

Reviewer 1 Report

The manuscript carried out a systematic study of the surface energy for statistical copolymers of different compositions and thermal prehistory, and evaluated the influence of the nature of the contacting surface and the functional groups of different polarity. The comments are listed below.

1.     Abstract: the authors should add the vital results to emphasize the innovation of this work.

2.     Introduction - The author should analyze the previous work with examples and draw out the significance of this work.

3.     The authors choosed the elastomeric materials based on statistical copolymers of styrene and butadiene, acrylonitrile and butadiene, butyl acrylate and vinyl acetate. Is there any basis for the selection?

4.     DSC test is an important method to analyze thermal properties. The authors should provide the DSC curves of copolymers and homopolymers to understand their thermodynamics features.

5.     It is better to give the notional chemistry structures for the different copolymers.

6.     It is necessary to compare the characteristics of statistical copolymers in this work and the reported work.

Author Response

Thank you for your review of our manuscript. We have answered each of your points below.

Point 1:  Abstract: the authors should add the vital results to emphasize the innovation of this work.

Response 1: The text of the abstract has been completed in accordance with the reviewer's comments and vital results have been indicated.

Point 2: Introduction - The author should analyze the previous work with examples and draw out the significance of this work.

Response 2: According to your recommendation, information was added to the text of the Introduction, emphasizing the place and importance of this work. It is also determined by the wide industrial use of the studied copolymers.

Point 3: The authors choosed the elastomeric materials based on statistical copolymers of styrene and butadiene, acrylonitrile and butadiene, butyl acrylate and vinyl acetate. Is there any basis for the selection?

Response 3: The reason for the choice of materials that are based on statistical copolymers of styrene and butadiene, acrylonitrile and butadiene, butyl acrylate and vinyl acetate is their wide use as part of adhesives and adhesives for various materials.

Point 4.     DSC test is an important method to analyze thermal properties. The authors should provide the DSC curves of copolymers and homopolymers to understand their thermodynamics features.

Response 4: We would like to point out that in the study we presented, the DSC data is given as a reference to discuss the results obtained, the DSC curves can be seen in the link [16].

Point 5.     It is better to give the notional chemistry structures for the different copolymers.

Response 5: We thank you for this wish. The conventional chemical formulas are not given in the text, because they are well known and, in our opinion, greatly increase the volume of the material.

Point 6.     It is necessary to compare the characteristics of statistical copolymers in this work and the reported work.

Response 6: We find this proposal very interesting.The work previously obtained by Wu on the example of statistical copolymers of ethylene oxide with propylene oxide [13] is mentioned in the text of the article. A comparison of the characteristics of polymers will be the subject of the next review publication, the main directions of which will be the experimental materials presented in this paper.

Reviewer 2 Report

The study of this work is to deepen the understanding of the state of interfacial boundary layers, the thermodynamics of polymer mixing and adhesive bonding strength. The research is correctly designed. The results are enough to draw conclusions with the evidence and arguments presented in the article.

The manuscript is presented in a structured manner. The introduction does provide background, but can’t be checked because many references are not available. The Objects and methods of investigation are described and the Results and discussion are commented but in some area the English writing is difficult to follow.

Recommendations for Authors

 Introduction:

-          Line 26: “In recent years, the synthesis of…… developed most intensively [5, 6].”

References 5 and 6 are from 2009, they should be from the last 5 years if the authors discus about the latest study.

-       References in Russian can’t be checked and I recommend for the authors to replace them with the English versions.

Objects and methods of investigation

-       are described with sufficient detail to allow understanding of the next section.

Results and discussion

-       the results are presented and interpreted in perspective of previous studies, the authors are explaining the results by citing previous work that can’t be verified (is in Russian), they should provide material in English.

-          the English writing is difficult to follow in some area,

-          where is Figure 3?

-          Line 175: “…for SNC γD...”  instead of SNC there should be BNC?

-          line 271: in Figure 6. plot (b) were is the line for PTFE (2)?

References

-          For all references that are in Russian 7-13,19, 22, 23, 25, 28-33 the authors should find an English source

-          Reference 14 could not be found online

 English language and style require minor spell checks

Line 128: „...elastic state.spaceNote that along...”

Line 265: “…structure of such layers is not equilibrium.”

Line 272: “Figure 6. Dependency γ (а) γ D (b) from….”

Line 318: “…air.spaceThe...”

Line 340: at reference 2 the 2 in front of “2Toussaint A. F. and Luner P….” should be removed

Author Response

Thank you for your review of our manuscript. We have answered each of your points below.

Point 1:  Recommendations for Authors.  Introduction:

-          Line 26: “In recent years, the synthesis of…… developed most intensively [5, 6].”

References 5 and 6 are from 2009, they should be from the last 5 years if the authors discus about the latest study.

-       References in Russian can’t be checked and I recommend for the authors to replace them with the English versions.

Response 1: Following the reviewer's comments, we have supplemented the abstract and added new sources to the list of references.

Point 2: Results and discussion

-       the results are presented and interpreted in perspective of previous studies, the authors are explaining the results by citing previous work that can’t be verified (is in Russian), they should provide material in English.

Response 2: The number of references in Russian has been significantly reduced.

Point 3:  - the English writing is difficult to follow in some area,

Response 3: We would like to thank you for your careful reading of the manuscript and your valuable comments. We have corrected the English style in all the places indicated by the reviewer in the manuscript.

Point 4:  - where is Figure 3?

Response 4: Figure 4 is included in the text of the article. It was originally present in the proposed manuscript. Its loss is due to a technical glitch.

Point 5:  -   Line 175: “…for SNC γD...”  instead of SNC there should be BNC?

Response 5: Corrected in accordance with the comment on BNC.

Point 6:  -  line 271: in Figure 6. plot (b) were is the line for PTFE (2)?

Response 6: An additional curve 2 for PTFE  has been added to Figure 6.

Point 7: References.   For all references that are in Russian 7-13,19, 22, 23, 25, 28-33 the authors should find an English source.

-          Reference 14 could not be found online

Response 7: We are pleased to inform you that the links have been substantially revised and changes have been made. Reference 14 has been replaced.

Point 8-12:  English language and style require minor spell checks

Line 128: „...elastic state.spaceNote that along...”

Response 8: Line 128: "...elastic state. Corrected.

Line 265: “…structure of such layers is not equilibrium.”

Response 9: Corrected to non-equilibrium.

Line 272: “Figure 6. Dependency γ (а) γ D (b) from….”

Response 10: Corrected to  Dependency

Line 318: “…air.spaceThe...”

Response 11: Line 318: “…air. The...”

Line 340: at reference 2 the 2 in front of “2Toussaint A. F. and Luner P….” should be removed

Response 12: Corrected, Line 340: reference 2 the 2 in front of “2Toussaint A. F. and Luner P….” is deleted.

Round 2

Reviewer 2 Report

Considering that the authors have made the requested changes I accept the manuscript in the present form to be published.